# Caffeine Placebo Effect in Sport and Exercise: A Systematic Review

**DOI:** 10.3390/nu16183219

**Published:** 2024-09-23

**Authors:** Alejandro Vega-Muñoz, Nicolás Contreras-Barraza, Guido Salazar-Sepúlveda, Nelson Lay, Miseldra Gil-Marín, Nicolás Muñoz-Urtubia

**Affiliations:** 1Facultad de Medicina y Ciencias de la Salud, Universidad Central de Chile, Santiago 8330507, Chile; alejandro.vega@ucentral.cl; 2Facultad de Ciencias Empresariales, Universidad Arturo Prat, Iquique 1110939, Chile; 3Pontificia Universidad Católica de Valparaíso, Valparaíso 2340025, Chile; 4Facultad de Ingeniería, Universidad Católica de la Santísima Concepción, Concepción 4090541, Chile; gsalazar@ucsc.cl; 5Facultad de Ingeniería y Negocios, Universidad de Las Américas, Concepción 4090940, Chile; 6Facultad de Educación y Ciencias Sociales, Universidad Andres Bello, Viña del Mar 2531015, Chile; nelson.lay@unab.cl; 7Public Policy Observatory, Universidad Autónoma de Chile, Santiago 7500912, Chile; miseldra.gil@uautonoma.cl; 8Instituto de Ciencias de la Educación, Universidad Austral de Chile, Valdivia 5090000, Chile; nicolas.munoz01@uach.cl; 9International Graduate School, University of Extremadura, 10003 Caceres, Spain

**Keywords:** caffeine, placebo, sport sciences, physical performance, psychophysiology, coffee studies

## Abstract

The objective of this review article is to systematically identify the caffeine placebo effect in sport and exercise activities. We selected randomized controlled trial studies to better understand the impact of caffeine and its placebo effect on sports performance. We extracted a set of articles that refer strictly to the topics of caffeine and its placebo effect in sport and exercise, considering the databases integrated to the Core Collection Web of Science and Scopus, as well as the registration of the documents in PubMed^®^, a system with a selection process aligned with the guidelines for the PRISMA methodology, establishing the eligibility criteria of the articles with the PICOS tool, to which a systematic review is applied. Finally, the results show that caffeine improves anaerobic capacity and endurance, while placebo perceived as caffeine can also increase performance by reducing pain and improving concentration. This finding underscores the influence of expectations and placebo on physical performance, suggesting that managing these expectations may be an effective strategy for improving athletic performance.

## 1. Introduction

The actual effectiveness of nutritional supplementation may be markedly influenced by the placebo effect; thus, many reported benefits may be due to perception rather than active intervention [1]. The expectation of having consumed caffeine may delay muscle fatigue, even without the actual presence of caffeine, underlining the influence of psychological expectations on physical performance [2]. Even the belief in caffeine intake can improve performance in recreational runners at speeds close to their aerobic maximum without altering pacing strategy, suggesting an advantage in Valero et al. [3].

Although placebo and nocebo effects usually have a small to moderate impact on sports performance, noticeable placebo aids can improve performance without the athlete perceiving additional effort [4]. High-relevance placebo ergogenic aids can enhance performance without the athlete’s awareness, and deceiving athletes appears unnecessary, as even believing one has received an inactive placebo supplement maintains the ergogenic response to the placebo [5]. Furthermore, placebos can reduce fatigue during the anticipatory phase during the movement, highlighting the role of the central nervous system [6]. Previous research even demonstrates that a coffee aroma can increase performance in reasoning tasks by raising expectations and physiological arousal, thus evidencing a placebo effect [7].

Thus, in the field of psychology, the following two main theories have been proposed to explain the placebo effect in people: the conditioning theory, which suggests that the placebo effect is a conditioned response, and the mentalistic theory, which holds that the patient’s expectation is the main basis for the placebo effect [8,9]. From the expectancy theory, it is observed that placebo effects are more likely to occur in participants with prior expectations, which are usually based on previous experiences or cultural beliefs [10]. Factors such as level of physical activity, salient memories of responses to exercise, and exposure to various sources of information influence the formation of these habitual expectations. In contrast, the so-called specific study expectations are more dynamic and can change throughout participation in research, integrating the experiences of the different phases of the study [11]. Recent studies have shown that expectation and belief can have a positive effect on individual perception, influencing confidence and belief change in people exposed to a placebo [12,13]. These studies, which are grounded in both conscious phenomenological approaches and the interaction from personality systems, indicate that mental systems, which have operations that do not need to be conscious, have the capacity to produce placebo effects. This phenomenon is due to implicit, parallel, and holistic networks that integrate aspects to the self and provide self-regulatory mechanisms [14].

Other studies have identified placebo-like effects in caffeine users, showing that both caffeine and its associated stimuli can influence patients’ perception of their willingness to engage in activities and cognition [15,16,17,18]. For example, performance was found to be lower in participants who believed caffeine would have the greatest benefit, highlighting a link between ergogenic expectancy, motivation, and personality characteristics [17]. Another investigation examined the effect of caffeine on contingent negative variation (CNV) and reaction time (RT) in 44 healthy, inexperienced volunteers. They found that caffeine increased early CNV amplitude in those with high scores and decreased it in those with low scores, supporting the hypothesis that caffeine might modulate CNV as a function of the individual’s basal activation level [19].

In addition, other research concluded that the effects that caffeine has on athletic performance may be due to psychological rather than physical benefits. It is suggested that coaches might use placebos to improve anaerobic performance when there are concerns about the side effects or cost of caffeine [16]. The benefits associated with caffeine consumption expectations appear to be more appropriate for tasks that require fewer specific cognitive attributes, but greater gross motor function. These effects may be greater when expectations are combined with caffeine pharmacology [20]. Still, data suggest that almost all individuals respond to caffeine, showing a positive response compared to a placebo in multiple tests [21], and regular coffee drinkers show physiological arousal in response to coffee-associated stimuli [22]. According to the meta-analysis of Wang et al. [23], differences in the ergogenic effects of caffeine in exercise can be explained by different genetic alleles.

Placebos affect a range of physiological processes (cardiovascular, gastrointestinal, and respiratory) through expectations for clinical improvement affecting heart rate, blood pressure, coronary diameter, gastric motility, intestinal motility, and pulmonary function [6,22]. Thus, in the field of physical performance, placebos can increase performance by increasing muscle work and decreasing perceived exertion [6]. Both a placebo and non-placebo can influence aspects of motor performance (speed, strength, and fatigue resistance), pain, and motor dominance [24]. Therefore, placebo effects from caffeine are subjective or behavioral beneficial outcomes from an intervention. These effects are not attributable to caffeine’s inherent properties but are derived from the expectations of individuals [25].

Thus, the neurobiological effects from caffeine translate into causal belief and the achievement of increased endurance and reduced fatigue. Caffeine acts centrally on the preparatory phase of the movement (readiness potential), something which emphasizes the role of the central nervous system in the generation for fatigue [6]. This readiness potential, as an electrophysiological sign of movement preparation, allows the detection of the reduction in the fatigue induced by the placebo; the increase in force resulting from the placebo is related to increased activity in the corticospinal system [24]. The results of another investigation showed that both caffeine and the placebo perceived as caffeine improved motor performance, despite no changes in motor cortex activation, and an observed increase in prefrontal cortex deoxygenation [26].

Even though, for Wicht et al. [25], caffeine-related expectations induce smaller effects than the substance itself, and its behavioral effect on sustained attention and inhibitory control is smaller, caffeine itself mainly influences cognitive processes and brain areas that support attention allocation. Other evidence in patients with Parkinson’s disease has shown that placebo effects on motor symptoms are related to changes in subcortical neuronal firing activity and dopamine release [24].

Specifically, in sport science studies, the influence of placebo and nocebo effects on sport performance is recognized. This influence is demonstrated in the research of Nasser et al. [27] in soccer, Wilson et al. [28] in marathon runners, Beedie et al. in cycling [29], Ferreira et al. [30] in sprint swimming, and Grgic [31] in vertical jump. However, the descriptive nature of most of these studies and the lack of more solid mechanistic evidence keeps placebo and nocebo effects in sport in an empirical gray area, giving way to skepticism, considering the effects as illusory, and attributing them to motivation. While neuroscientific research has identified neurobiological and physiological interactions associated with placebo and nocebo effects, providing data for these effects is an area of potential interest for sport scientists [32]. 

This lack of clarity motivates us to propose as an objective of this study, to systematically review research with a quantitative randomized controlled trials design in the search for solid conclusions regarding the placebo effect of caffeine in sport and exercise. To this end, it seems essential to distinguish between the concepts of physical activity, physical exercise, and sport. Physical activity is defined as any bodily movement that produces energy expenditure, ranging from daily tasks to recreational and occupational activities [33,34]. On the other hand, physical exercise is a subcategory of physical activity that is characterized by being planned, structured, and repetitive, with the primary goal of improving or maintaining one or more components of physical fitness [33,34]. In contrast, sport is a form of exercise that, in addition to sharing these characteristics, is generally performed competitively, either individually or in teams, and is governed by specific rules. Sport is also organized according to factors such as age, gender, weight, or skill level, with clearly defined objectives for participants or teams [34,35].

## 2. Materials and Methods

In this review, the Preferred Reporting Items for Systematic Reviews and Meta-Analyses (PRISMA) guidelines [36] were used, and the PICOS (participants, interventions, comparators, outcomes, and study design) strategy was used to establish the eligibility criteria for the articles [37]. The registered protocol is in PROSPERO (ID589775).

According to the current checklist of the PRISMA guidelines [36], the following quality steps for systematic reviews were verified according to the following items: (1) title, (2) structured abstract, (3) rationale, (4) objectives, (5) eligibility criteria, (6) sources of information, (7) search strategy, (8) selection process, (9) data extraction process, (10a) and (10b) data items, (11) study risk of bias assessment, (13) methods of synthesis, (14) reporting bias assessment, (16a) and (16b) study selection, (17) study characteristics, (18) risk of bias in studies, (19) results of individual studies, (20) results of syntheses, (23) discussion, (24) registration and protocol, (25) support, (26) competing interests, and (27) availability of data, code, and other materials. The following items were excluded from the PRISMA guidelines due to their non-applicability to the objectives of this review: (12) effect measures, (15) certainty assessment, (21) reporting biases, (22) certainty of evidence. In addition, the initial search for articles was performed using bibliometric procedures [38].

### 2.1. Search Strategy

We used a set of articles from two databases, with equivalent search vectors and without additional restrictions (such as period of years or types of documents), extracted from Web of Science—Core Collection (WoSCC) [39] and Scopus [40], selecting articles published in journals indexed in these databases based on the following search vector on WoS: TS = ((Placebo NEAR/0 Effect) AND (caffeine)). The following search vector was used on Scopus: TITLE-ABS-KEY ((placebo W/0 effect) AND (caffeine)). The TS of WoS makes a simultaneous search in the fields title, abstract, and keywords (author and plus); the TITLE-ABS-KEY of Scopus makes a simultaneous search in the fields title, abstract, and keywords (author and Index), and the NEAR and W are nearness operators.

### 2.2. Eligibility Criteria

The selection of articles was specified based on the following eligibility criteria: the target population (participants), interventions (methodological techniques), elements of comparison of these studies, outcomes of these studies, and study designs (the criteria of the PICOS strategy as shown in Table 1).

### 2.3. Study Selection and Data Extraction

In the first step, duplicates were manually removed. Next, only the documents that WoS and Scopus declared as indexed in PubMed^®^ (Bethesda, MD, USA) [42] were selected and reconstructed in this database by screening according to their data and metadata filters, limiting the inclusion to the following: Article type: Randomized Controlled Trial; Species: Humans; Sex: Female and Male; and Age: Adult: 19–44 years (The young adult population was taken, according to the PubMed^®^ age range, to reduce the presence of comorbidity in the subjects studied). The titles and abstracts of the papers were then checked for relevance by two investigators who independently reviewed the full texts of potentially eligible papers, applying the inclusion criteria (See Table 1). Any disagreement was discussed with a third investigator until a consensus was reached.

### 2.4. Quality Assessment and Risk of Bias

The MMAT (Mixed Methods Appraisal Tool) scale was used to assess the risk of bias, specifically its methodological quality measure, as follows: screening questions (for all types) and quantitative randomized controlled trials with focus in sport and exercise. Two authors conducted the studies independently, and a third author was included in case of any discussion.

The Mixed Methods Appraisal Tool (MMAT) checklist used measures 7 items (2 screening questions, and 5 quantitative randomized controlled trials) according to a score from zero to one (which we used dichotomously), to obtain a final percentage mean. Studies are considered high quality > 75%, moderate quality 50–74% and low quality < 49%. Studies with values below 75% were excluded from the analysis and discussion by categories [43].

### 2.5. Categories Proposed

In this paper, we have analyzed the ergogenic effect of caffeine aid and the placebo effect. For the placebo effect we have used the categories proposed by Beedie et al. [29] in the following four categories: pain reduction, belief–behavior relationship, attentional changes, and changes in arousal.

## 3. Results

The bibliometrics search of articles identified a total of 174 articles from two databases (Web of Science Core Collection and Scopus, see in Appendix A). There were 174 unique titles and abstracts (without repetition); however, 29 articles were excluded that did not have the PubMed^®^ code which reduced the corpus analyzed to 145 documents in Appendix B (Documents retrievable from PubMed^®^ by PMID code). 

The studies were retrieved and screened using the selection criteria defined with the PICOS tool as shown in Figure 1.

The exclusions corresponded to articles that did not refer to the 26 selected papers that met the criteria. We performed a review of the complete paper to determine if it was related to physical activity. We discarded 14 of them. In addition, one of them did not make it to the paper. As a result, 15 studies were discarded (See Table 2).

Table 2 includes the identification of the eleven studies selected to review the effect of caffeine in an ergogenic manner and as a placebo effect. In the table, we indicate the PubMed^®^ identification number, the first author, the year of publication in the journal, and the respective DOI.

### 3.1. Eligibility Criteria Using Mixed Methods Appraisal Tool (MMAT)

Applying the MMAT criteria to the quantitative randomized clinical trials section, all eleven studies were found to meet the criteria of randomization, blinding, adherence to the intervention, lost to follow-up, and quality of the results. Therefore, they can be adequately considered in this study (Appendix C, Table A1).

Table 3 shows that of the total number of studies selected, all except for Beedie [29] (with four categories) and Filip-Stachnik [51] (with zero categories) are present in two categories. Regarding the dimensions with the most studies, this is led by Belief–behavior relation with 8 studies, followed by Arousal changes with 8, and Attentional changes with 5 studies. The “x” in the table means that the research shows some kind of effect in the proposed categories; if “x” does not appear, it means that there is no effect in those categories.

### 3.2. Results Caffeine Ergogenic Aid

The pharmacological effects that caffeine has on athletic performance have been consistently demonstrated in the studies reviewed, showing significant improvements in anaerobic capacity and endurance. In the study by Anderson et al. [16], ingestion with 280 mg of caffeine resulted in a marked increase in peak power generated during sprint tests in trained cyclists, evidenced by an improvement in speed to peak power and greater sustained power. Similarly, Saunders et al. [45] found that caffeine supplementation at a dose of 6 mg/kg increased performance in a cycling time trial, with a 4.1% increase compared to placebo, reinforcing caffeine’s ability to improve aerobic and anaerobic performance. Furthermore, Beedie et al. [29] demonstrated that caffeine, even in moderate doses, can significantly increase power output in cyclists during stress tests, underscoring its efficacy as an ergogenic agent to optimize sports performance in high-intensity activities. These findings highlight the robustness of the pharmacological effect of caffeine in improving both power and endurance, confirming its usefulness in the sports context.

### 3.3. Results Pain Reduction

In the study by Beedie et al. [29], the placebo effect of caffeine on cycling performance was explored, specifically highlighting pain reduction as a key mechanism. Participants who believed they had ingested different doses of caffeine consistently reported experiencing less pain during exercise, allowing them to maintain a more sustained effort. This perception of pain reduction was particularly pronounced in those who believed they had consumed a high dose of caffeine, even though they only received a placebo. Subjects reported phrases such as “the pain disappeared” and “I was able to exert myself more with less pain”, suggesting that the mere belief in the ingestion of an ergogenic agent can modulate the experience of pain and, consequently, improve physical performance. These findings underscore the power of placebo effect in pain management during exercise and its potential to positively influence athletic performance.

### 3.4. Results of the Belief–Behavior Relationship

In the study by Beedie et al. [29], it was observed that cyclists who believed they had ingested caffeine increased their average power output by 3.1% during cycling trials, despite receiving a placebo, demonstrating a clear dose-response effect based solely on belief. Similarly, Anderson et al. [16] found that cyclists who incorrectly believed they had consumed caffeine reached their peak power faster and experienced a greater drop in power, underscoring the significant impact that belief has on anaerobic performance. In other research, Saunders et al. [45] reported that those participants who correctly identified caffeine intake significantly improved their performance by 6.5%, while those who mistakenly believed they had ingested a placebo also showed improvements, albeit smaller, highlighting the influence of the placebo effect on physical performance. Furthermore, Duncan et al. [50] noted that the perception of having consumed a substance that could improve performance was sufficient for subjects to complete a greater number of repetitions in endurance exercise compared to control conditions, emphasizing the influence the belief has on behavior. On the other hand, Hurst et al. [49] found that athletes who intended to use sports supplements showed a greater placebo response, improving their performance in repeated sprint tests compared to those who did not intend to use them. These studies highlight that the relationship between belief and behavior is a determining factor in the placebo response, suggesting that athletes’ expectations can be manipulated to effectively improve their performance.

### 3.5. Results of Attentional Changes

In the study by Beedie et al. [29], it was observed that cyclists who believed they had consumed caffeine, but received a placebo, reported being more focused on their performance, which allowed them to divert their attention away from pain and perceived exertion, thus improving the performance of power output, blood lactate, oxygen and heart rate. In another study by Costa et al. [47], paralympic athletes who believed they had consumed caffeine showed a significant improvement in average velocity during bench throwing events, attributed to an attentional shift that allowed them to focus more on the explosive movement. In the study by Saunders et al. [45], it was evident that cyclists who believed they had ingested caffeine, despite receiving a placebo, maintained greater concentration and mental endurance during endurance tests, allowing them to prolong time to fatigue compared to those who did not believe they had received any supplementation. Similarly, the Duncan et al. [50] study found that participants who believed they had received an ergogenic supplement exhibited greater focus on their performance during strength exercises, which translated into an increase in the number of repetitions completed.

### 3.6. Result of Arousal Changes

Placebo effects related to changes in arousal have been extensively studied, showing how the belief of having ingested an ergogenic substance can significantly modify alertness and physical performance. In the study by Duncan et al. [50], it was observed that participants who believed they had consumed caffeine experienced an increase in motivation and a higher level of arousal, which allowed them to complete more repetitions in endurance exercises compared to those who believed they had ingested a placebo. Similarly, Anderson et al. [16] found that cyclists who incorrectly believed they had ingested caffeine showed a decrease in time to peak power and a greater drop in power, suggesting that belief in having consumed a stimulant increases arousal and, consequently, anaerobic performance. On the other hand, Beedie et al. [29] observed that subjects who received a placebo under the belief that it was caffeine reported typical caffeine symptoms, such as increased energy and concentration, suggesting that placebo-induced arousal may simulate the actual effects of caffeine. Finally, in the study by Costa et al. [47], paralympic athletes who believed they had ingested caffeine showed an improvement in mean velocity in explosive movements, attributed to a placebo-induced increase in mental and physical arousal.

## 4. Discussion

The systematic review presented in this manuscript included 174 scientific articles from two databases, Web of Science and Scopus. Quantitative studies with randomized clinical trials (under PubMed standard) selection criteria resulted in 11 articles reviewed in depth, which allowed an analysis based on five categories related to how the ergogenic effect of caffeine helps reduce pain and can have a belief–behavior relationship, attentional changes, and changes in arousal.

Regarding the first category, caffeine as an ergogenic aid, this systematic review agrees with other previous reviews in its effect on sports and exercise performance on the human body mainly through the following five mechanisms: Antagonism of adenosine, increased fatty acid oxidation, nonselective competitive inhibitor of the phosphodiesterase enzymes, Increased post-exercise muscle glycogen accumulation and Mobilization of intracellular calcium [52]. These mechanisms improve the reaction time and agility [53] in sports and exercise based on endurance, resistance, and aerobic and anaerobic activities [54]. The studies coincide with the effects of caffeine and its incidence in aspects of fat oxidation rate during exercise [44], as well as in sports performance resulting from the intake of a caffeine dose [16]. These findings lead us to the fact that of all the rigorous studies that comply with the methodological requirements proposed in this review, only three have carried out experiments relating to direct caffeine intake in conjunction with corresponding placebos, evidencing an important gap for future lines for research.

In the measurement for placebos, in the category called Pain reduction, only one study clearly refers to the effects of caffeine used as a placebo [29], and measured and reported in a qualitative manner on the perception of the six participating subjects. It is the category with the fewest reports, so it could also be important to carry out studies to support or rule out this topic as a relevant category. One of the most reported categories in this review is the Belief–behavior relationship, corresponding to studies indicating placebo effects affecting power performance [29], anaerobic performance [16], physical training [45], endurance [50], and sprint [49].

On the other hand, the Attentional changes category shows that across studies (all studies in the category), the level of attention and concentration can deflect pain [29] as well as result in increased focus, increased speed [47], mental endurance [45], and focus affecting strength [50]. Finally, the last category of placebo effects are Arousal changes, where studies about activation showed effects on motivation [50] and performance [16,29,47]. While these categories proposed by Beddie allowed us to analyze the placebo effects of caffeine on sports performance, it also allowed us to point out the shortcomings in the Quantitative randomized clinical trials studies with respect to the designs in the placebo sections. Many of these studies lack adequate methodologies to mitigate the effect biases that may be part of the environment or the experiences of the participants themselves.

Therefore, considering the results, it is necessary to generate a more exhaustive model in the research designs for the application on placebos that will allow a better parameterization for the biases inherent to studies in human subjects. In line with the “room for improvement” topic raised by Halperin et al. [55], there is a need to examine the reliability and validity of research practices in exercise and sport sciences. Building on the literature review, it is essential to accurately identify the psychological effects influenced by both expectation theory and conditioning theory. A more nuanced understanding of the subcategories within expectation theories could enable a more precise analysis of their effects. Similarly, in the case of conditioning theory, breaking the theory down into subcategories would allow for more rigorous and segmented analyses. Additionally, aligning the analysis with categorized levels of influence on the effects would further enhance the precision of the findings.

The sustainability and long-term effects of using placebos in sport remain an open question. While the immediate benefits of placebo effects are evident, the durability of these effects over time, particularly once athletes become aware of the placebo nature of the intervention, is less clear. Research indicates that placebo effects might diminish if athletes realize they have not consumed an actual ergogenic aid [29]. This issue raises important considerations about the long-term viability of using placebos as a consistent performance-enhancing strategy and suggests that the timing and context of such interventions are crucial. Ethical considerations also play a significant role in the discussion of placebo use in sports. The deliberate use of placebos raises questions about the ethics of deceiving athletes, even if the intention is to enhance their performance [22]. Coaches and sports organizations must balance the potential benefits of placebo use with the moral implications of such strategies, potentially exploring ways to achieve similar psychological effects without deception, such as through informed placebo use or enhancing the psychological readiness of athletes. Finally, the implications for anti-doping policies are substantial. If psychological interventions like placebo use can produce performance enhancements comparable to banned substances, it challenges the current framework of anti-doping regulations. The findings from this review suggest a need for a broader understanding of performance enhancement, one that includes psychological and placebo-induced effects, alongside traditional pharmacological considerations [32]. The findings of our research, both in ethical aspects and anti-doping policies, may influence the generation of unregulated and regulated protocols in the transparent declaration of the use of placebos in high-performance competitions, contributing to the generation of a reinforcement of the culture of fair competition behavior. Future research should focus on integrating these psychological aspects into the overall approach to athlete performance and regulation in competitive sports.

## 5. Conclusions

This study evaluated the ergogenic effect of caffeine and the impact of placebo effect on sports and exercise, using the following Beedie et al. categories: pain reduction, belief–behavioral relationship, attentional changes, and alterations in arousal in studies with reported RCTs.

The results of our research confirm that caffeine consistently improves exercise and sport performance based on endurance, resistance, and aerobic and anaerobic activities, while placebo, when perceived as caffeine, can also significantly improve performance. More evidence is needed, and the topic needs to be further explored. Our results showed that the studies related to the placebo effect of caffeine in exercise and sport have mostly presented its effects on the Belief–Behavior Relationship and Changes in arousal; however, the reduction in pain is present in only two studies. This evidence is one of the main limitations of this research in our investigation, which is the scarce scientific productivity on the findings of placebo effects in exercise and sport using the RCT methodology, which allows us to invite the scientific community to continue exploring this field of study. Throughout the development of our research, we found it crucial to establish clear conceptual definitions of sport and exercise to accurately classify the RCT studies. To achieve this, we referenced the studies of Caspersen et al. [33], Sancassiani et al. [34], and Malm et al. [35], which ensured consistency in our work and enabled us to correctly distinguish exercise and sport activities in relation to their respective populations, as presented in Table 3.

Finally, we conclude that there is great potential for development in the aspects of the placebo effect of caffeine in sport and exercise. We encourage the increase in randomized controlled trials on this topic. Future research could focus on finding ways to clearly systematize the findings in physiological, psychological and psychophysiological aspects.

## Figures and Tables

**Figure 1 nutrients-16-03219-f001:**
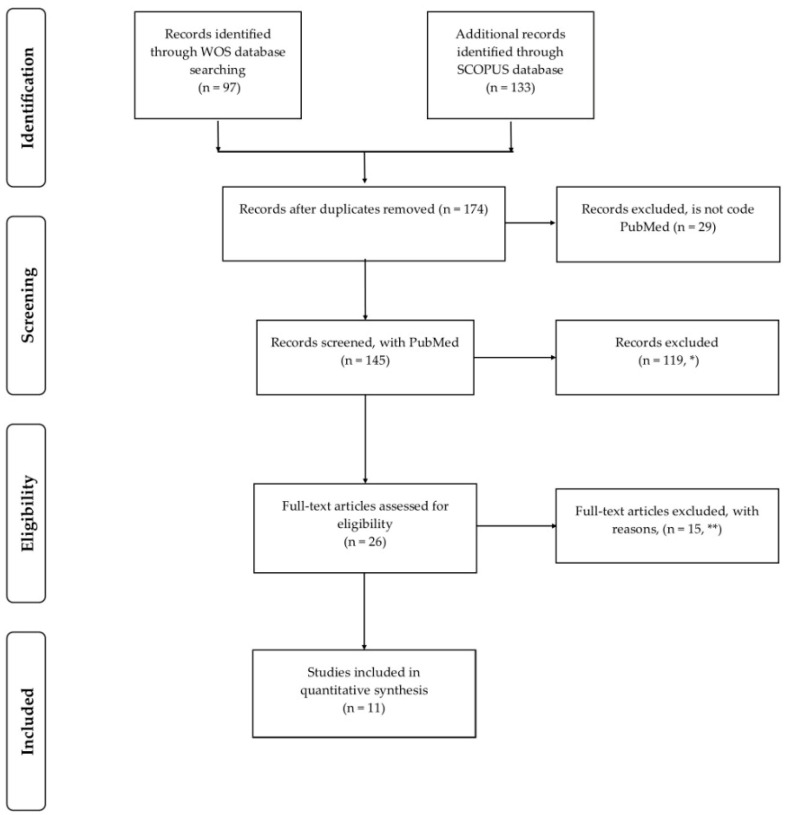
Preferred Reporting Items for Systematic Reviews and Meta-Analyses (PRISMA) analysis flow. * The exclusions corresponded to articles that did not refer to Randomized Controlled Trial, Species: Humans, Sex: Female, and Male, and Age: Adult: 19–44 years. ** It was not possible to access one full-length article, and fourteen articles did not correspond to the subject.

**Table 1 nutrients-16-03219-t001:** Eligibility criteria using PICOS (participants, interventions, comparators, outcomes, and study design).

PICOS	Description
Population	Persons (humans), with age restriction between 19 and 44 years old (adults), who practice sports or exercise.
Interventions	Application of experiments, questionnaires, interviews, or health and motor physical exams in humans.
Comparator	Placebo effects of caffeine reported according to the categories of Beedie et al. [29].
Outcomes	Placebo effects of caffeine reported in each study.
Study designs	Randomized Controlled Trial (RCT) type was included (under MMAT quality criteria) [41].

**Table 2 nutrients-16-03219-t002:** Set of 11 articles selected by the PRISMA protocol.

PubMed Identifier (PMID)	First Author	Pub. Year	Journal	Digital Object Identifier (DOI)
33673567	Gutiérrez-Hellín J [44]	2021	Nutrients	https://doi.org/10.3390/nu13030782
27882605	Saunders B J [45]	2017	Scand J Med Sci Sports	https://doi.org/10.1111/sms.12793
38794643	Ortiz-Sánchez D [46]	2024	Nutrients	https://doi.org/10.3390/nu16101405
30782172	Costa GCT [47]	2019	J Int Soc Sports Nutr	https://doi.org/10.1186/s12970-019-0276-9
25412293	Ross R [48]	2015	Med Sci Sports Exerc	https://doi.org/10.1249/MSS.0000000000000584
17146324	Beedie CJ [29]	2006	Med Sci Sports Exerc	https://doi.org/10.1249/01.mss.0000233805.56315.a9
31996613	Anderson DE [16]	2020	J Strength Cond Res	https://doi.org/10.1519/JSC.0000000000003537
28419027	Hurst P [49]	2017	Med Sci Sports Exerc	https://doi.org/10.1249/MSS.0000000000001297
19567927	Duncan MJ [50]	2009	Int J Sports Physiol Perform	https://doi.org/10.1123/ijspp.4.2.244
29889868	Broelz EK [5]	2018	PLoS One	https://doi.org/10.1371/journal.pone.0198388
33322129	Filip-Stachnik A [51]	2020	Nutrients	https://doi.org/10.3390/nu12123813

**Table 3 nutrients-16-03219-t003:** Categories caffeine and placebo caffeine effect in sport and exercise activities.

Study (RCT)	Caffeine Effect (Ergogenic Aid)	Placebo Effects Categories:					Count Placebo Effect
		Pain Reduction	Belief–Behavior Relation	Attentional Changes	Arousal Changes	Practice Population	
Gutiérrez-Hellín J [44]	X			X	X	Exercise	2
Saunders B [45]	X		X	X		Exercise	2
Ortiz-Sánchez D [46]			X	X		Exercise	2
Costa GCT [47]				X	X	Sport	2
Ross R [48]			X		X	Sport	2
Beedie CJ [29]		X	X	X	X	Sport	4
Anderson DE [16]	X		X		X	Sport	2
Hurst P [49]			X		X	Sport	2
Duncan MJ [50]			X		X	Exercise	2
Broelz EK [5]			X		X	Sport	2
Filip-Stachnik A [51]						Exercise	0
Count Effects:	3	1	8	5	8		

X: The “x” in the table means that the research shows some kind of effect in the proposed categories; if “x” does not appear, it means that there is no effect in those categories.

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
