# Peer review of "Caffeine Placebo Effect in Sport and Exercise: A Systematic Review"

_nutrients, 2024, doi:10.3390/nu16183219_

Round 1

Reviewer 1 Report

Comments and Suggestions for Authors

Dear Corresponding Author,
thank you for submitting your article to Nutrients and congratulations on your work.

Brief Summary
This study presents a systematic review of the placebo effect of caffeine in sports activities. The main objective was to examine the impact of caffeine and its placebo effect on sports performance through the analysis of randomized controlled trials. The methodology followed the PRISMA guidelines, using the MMAT to assess the quality of included studies. The main results highlight that caffeine improves anaerobic capacity and endurance, while placebo perceived as caffeine can increase performance by reducing pain and improving concentration. The study underlines the influence of expectations and the placebo effect on physical performance, suggesting that managing these expectations could be an effective strategy to improve athletic performance.

General Comments
The work addresses a topic of great relevance in the field of sports science and nutrition, offering an updated and comprehensive synthesis of existing literature on the placebo effect of caffeine. The methodological approach is generally solid, with appropriate use of PRISMA guidelines and MMAT for quality assessment of studies.

However, there are some aspects that in my opinion could be improved to further increase the quality and impact of the work, and which the authors can consider whether to apply or not:

  • The introduction could benefit from a broader contextualization of the importance of the placebo effect in sport, perhaps including specific examples of how this phenomenon has been observed in various sports disciplines.
  • The methods section, although generally well-structured, could be enriched with more details on the search strategies used and the inclusion/exclusion criteria for studies.
  • The presentation of results could be improved through the use of summary tables or graphs that allow for a more immediate visualization of the main findings.
  • The discussion could be expanded to explore more deeply the practical implications of these results for athletes, coaches, and sports professionals.

Specific Comments

  • Lines 47-57: This section on psychological theories of the placebo effect is interesting, but could be linked more explicitly to the sports context. Perhaps you could add an example of how these theories specifically apply to athletic peformance? Is there evidence or is there a lack of evidence and are yours hypotheses?
  • Lines 101-110: The paragraph discussing the lack of solid mechanistic evidence is crucial. It might be useful to elaborate further on what types of studies would be needed to fill this gap in the literature?
  • Line 111: The Systematic review does not seem to have been registered on Prospero, can you specify the reason?
  • Line 129: I appreciate the methodology in general, however it would be necessary to integrate (also with an attached table) the original query used for each database to allow perfect reproducibility, at the moment it is not possible.
  • Line 145, Table 1: The PICOS table is very useful, but the specified age (19-44 years) seems arbitrary. Could you briefly explain why you chose this age range?
  • Lines 186-190: The study selection process could be described more clearly. For example, could you specify how many reviewers independently examined the titles and abstracts? It's also not clear in the final "Author Contributions" section.
  • Lines 201-204, Table 3: This table is informative, but could be improved by adding a brief legend explaining the meaning of the "X"s and numbers in the "Total Placebo Effect" column.
  • Lines 285-291: In the discussion, you mention the need for a more comprehensive model for placebo effect research. Do you have any specific suggestions on how this model could be structured?
  • Lines 329-341: This section on ethical implications and anti-doping policies is very interesting. Could you expand a bit on how your findings might influence future policies in this area?
  • Line 342: While I find the study interesting, I believe there are some relevant limitations to be integrated, but there is no information about this.

In general, the manuscript is well written, but there are some minor typographical and grammatical errors that should be corrected. I'm not referring to the quality of the English where I don't consider myself an expert, but to broader errors. For example:

  • Line 38: "an advantage in" seems incomplete. Perhaps you meant "an advantage in performance"?
  • Line 78: "a number on" should be "a number of"

Also, some sentences are a bit long and complex, which can make understanding difficult. Consider breaking up some of these sentences to improve readability.

Despite these observations, your work represents a significant contribution to the literature on the placebo effect in sport. With some minor revisions, I believe this article will be of great interest to Nutrients readers and the scientific community in general.

I wish you good work on the revision and look forward to seeing the final version of your manuscript.
Kind regarsd

Author Response

We thank you for your comments, which have helped us to improve our research. We have incorporated them by highlighting them in green for a better understanding. 

C1: The introduction could benefit from a broader contextualization of the importance of the placebo effect in sport, perhaps including specific examples of how this phenomenon has been observed in various sports disciplines.

R: We have made it explicit with some examples of soccer, marathon, cycling, sprint swimming and vertical jumping. (Lines 118-121).

C2; The methods section, although generally well-structured, could be enriched with more details on the search strategies used and the inclusion/exclusion criteria for studies.

R: We have generated some changes in table 1 in sections 2.3 and 2.4.

C3; The presentation of results could be improved through the use of summary tables or graphs that allow for a more immediate visualization of the main findings.

R: We have made changes to section 3.1, in addition to improving table 3.

C4; The discussion could be expanded to explore more deeply the practical implications of these results for athletes, coaches, and sports professionals.

R: in the discussion we have expanded on some of the more specific aspects by focusing on policies and protocols. 

C5; Lines 47-57: This section on psychological theories of the placebo effect is interesting, but could be linked more explicitly to the sports context. Perhaps you could add an example of how these theories specifically apply to athletic peformance? Is there evidence or is there a lack of evidence and are yours hypotheses?

R: We have incorporated a better explanation of  the expectancy theory and its relationship to placebo effects. (Lines 56 - 63)

C6; Lines 101-110: The paragraph discussing the lack of solid mechanistic evidence is crucial. It might be useful to elaborate further on what types of studies would be needed to fill this gap in the literature?

R: We have incorporated some future lines of research and possible areas for development, both in the discussion (Line376-378) and in the conclusions (Lines 398-402)

C7; Line 111: The Systematic review does not seem to have been registered on Prospero, can you specify the reason?

R: We have already done so, we have the request under ID589775, we appreciate your comment to clarify it in writing. (Lines 133-134)

C8; Line 129: I appreciate the methodology in general, however it would be necessary to integrate (also with an attached table) the original query used for each database to allow perfect reproducibility, at the moment it is not possible.

R: The query vector for wos was reported on line 152 TS=((Placebo NEAR/0 Effect) AND (caffeine)), for Scopus it was reported on line 152-153 TITLE-ABS-KEY ((placebo W/0 effect) AND (caffeine)). Additionally we have added supplementary material with both databases.

C9; Line 145, Table 1: The PICOS table is very useful, but the specified age (19-44 years) seems arbitrary. Could you briefly explain why you chose this age range?

R: The Young Adult population was taken, according to the Pubmed age range to reduce the presence of comorbidity in the subjects studied. 

C10; Lines 186-190: The study selection process could be described more clearly. For example, could you specify how many reviewers independently examined the titles and abstracts? It's also not clear in the final "Author Contributions" section.

R: This was contained in the article, but we have highlighted it in green in lines 169 to 172, and 176 to 178.

C11; Lines 285-291: In the discussion, you mention the need for a more comprehensive model for placebo effect research. Do you have any specific suggestions on how this model could be structured?

R: We have expanded on the suggestions already made in the discussion. (Lines 314 to 319)

C12; Lines 329-341: This section on ethical implications and anti-doping policies is very interesting. Could you expand a bit on how your findings might influence future policies in this area?

R: We have expanded on the suggestions already made in the discussion. (Lines 372 to 378) 

C13; Line 342: While I find the study interesting, I believe there are some relevant limitations to be integrated, but there is no information about this.

R: One limitation is the paucity of RCT studies on placebo effects in sport and exercise, which we have reinforced in the discussion. (Lines 389-392)

C14; Line 38: "an advantage in" seems incomplete. Perhaps you meant "an advantage in performance"?

R: We have incorporated the author Valero in line 42.

C15; Line 78: "a number on" should be "a number of"

R: we have corrected the above.

We appreciate your review.

Reviewer 2 Report

Comments and Suggestions for Authors

The abstract needs some revision to avoid repeat, e.g. there 3x mention of systematic review.

Sport can be defined as “an activity involving physical exertion and skill in which an individual or team competes against another or others for entertainment.” Were the studies examined for the review on effects of caffeine/placebo in Sport.

Did you miss some studies, e.g. doi: 10.1123/ijspp.2019-1028 and doi: 10.3389/fphys.2018.01144.

L42. “they may still experience”. Please revise.

L74. “Still, data suggest that almost all individuals respond to caffeine”. There are many studies reporting on non responders linked to genotype, e.g. see doi: 10.1016/j.jshs.2023.12.005. I suggest to provide some information on the genotype effects of caffeine.

The conclusion is vague. Please be specific. Please revise the conclusion.

L78. Please reconsider “pathological”. Is there evidence for that part of the statement.  

Table 1 does not mention sport. Should the intervention be an intervention for examining sport performance. It seems your review is examining placebo effects in exercise studies and not sports studies. The term “sport” does not appear in the methods section.

Some and maybe all of the selected studies are not on sport, Gutiérrez-Hellín J [32] is on substrate oxidation during exercise. The authors should reconsider use of the term “sport”. In addition, in L188 there is now mention of “physical activity”. The authors should provide descriptions of what is considered sport, exercise and physical activity as they seems to use analogous terms for phenomena that are different.

Ls 194-195. Why PMID and DOI numbers. Do we need two?

Table 3. As an example. Is pain reduction a “sport activity”. Please revise the Table legend.

L257. The authors should avoid phrases like “thus improving their performance”. Performance can be so many things. The manuscript needs be clear on the details on the outcome parameters. Anytime “performance” is being mentioned, please be specific.

L342. According to the information in the conclusion the focus seem to have been on psychological factors important for exercise tasks. The manuscript needs focus and correct us of terminology.

L360. Please delete “Authorship must be limited to those who have contributed substantially to the work reported.”

Appendix A does not make sense. Please clarify.

Author Response

We thank you for your comments, which have helped us to improve our research. We have incorporated them by highlighting them in green for a better understanding. 

C1; Did you miss some studies, e.g. doi: 10.1123/ijspp.2019-1028 and doi: 10.3389/fphys.2018.01144.

R: These studies have been incorporated into the manuscript, (Line110 and 120) but are not part of the review since they do not contain the exact phrase “placebo effect” in the title, abstract or keywords. 

C2; L42. “they may still experience”. Please revise.

R: We have changed the redaction, thank you for your comment. 

C3; L74. “Still, data suggest that almost all individuals respond to caffeine”. There are many studies reporting on non responders linked to genotype, e.g. see doi: 10.1016/j.jshs.2023.12.005. I suggest to provide some information on the genotype effects of caffeine.

R: We have added a new reference (Line 88-91). We appreciate your comments. 

C4: The conclusion is vague. Please be specific. Please revise the conclusion.

R: We have expanded the conclusion. 

C5; L78. Please reconsider “pathological”. Is there evidence for that part of the statement. 

R: We have changed the concept to physiological processes (Line 92). thank you for your comment. 

C6; Table 1 Does not mention sport. Should the intervention be an intervention for examining sport performance. It seems your review is examining placebo effects in exercise studies and not sports studies. The term “sport” does not appear in the methods section.

R: In Table 1 We have incorporated the sport and exercise in the PICOS method. We have also added sport and exercise (Line 176).

C7; Some and maybe all of the selected studies are not on sport, Gutiérrez-Hellín J [32] is on substrate oxidation during exercise. The authors should reconsider use of the term “sport”. In addition, in L188 there is now mention of “physical activity”. The authors should provide descriptions of what is considered sport, exercise and physical activity as they seems to use analogous terms for phenomena that are different.

R: We have better clarified the constructs of our study of exercise and sport for a better understanding of the research. We consider this to be an important point and have highlighted it in the conclusions by ascribing to the definition of Carpersen et al. We also modified the title and clarified the method in Table 1.

C8; Ls 194-195. Why PMID and DOI numbers. Do we need two?

R:We have improved Table 2. Not all wos articles have a doi. Articles that have a doi are not necessarily registered in PubMed (Some do not have a PMID), on the contrary articles registered by a PMID do not always have a doi. Our eleven articles have both and that is an exception. The retrieval of documents in PubMed requires a PMID. 

C9; Table 3. As an example. Is pain reduction a “sport activity”. Please revise the Table legend

R: Thank you very much, we have improved table 3. 

C10; L257. The authors should avoid phrases like “thus improving their performance”. Performance can be so many things. The manuscript needs be clear on the details on the outcome parameters. Anytime “performance” is being mentioned, please be specific.

R: We have specified the performance (Line 278-279). We appreciate your comment. 

C11; L342. According to the information in the conclusion the focus seem to have been on psychological factors important for exercise tasks. The manuscript needs focus and correct us of terminology.

R: We have expanded our findings in order to better communicate our conclusive ideas. We have added references that specify the required use of the terminology (Line 395).

C12; L360. Please delete “Authorship must be limited to those who have contributed substantially to the work reported.”

R: We have removed the phrase.

Round 2

Reviewer 1 Report

Comments and Suggestions for Authors

Dear authors, I have carefully read the latest version of the manuscript, and I appreciate the changes. I believe that this version is more consistent, and I have no hesitation in confirming my personal interest in publishing this article in its current form.

Author Response

C1. Dear authors, I have carefully read the latest version of the manuscript, and I appreciate the changes. I believe that this version is more consistent, and I have no hesitation in confirming my personal interest in publishing this article in its current form.

R1. Dear Reviewer, we appreciate your comments, which have greatly improved our manuscript. We highlight in light green, the changes requested by the review team.

Reviewer 2 Report

Comments and Suggestions for Authors

Thanks for your replies. There is still the issue of the absence of clear descriptions. See my comment below.

In your response to our comment about sport etc, you mention Carpersen et al. There is a reference to Casperen et al. That reference does not provide a definition/description of sport. Descriptions of what is considered sport, exercise and physical activity is still missing. Please be clear in your review what is considered sport, exercise and physical activity. In L392 is stated “Throughout the development of our research, we found it crucial to establish clear conceptual definitions of sport and exercise to accurately classify the RCT studies.” However, the reader will not find the “clear conceptual definitions” in your review.

Table 2. Are there books listed in the Table? Please reconsider the heading “journal/books”.

Author Response

C1: Thanks for your replies. There is still the issue of the absence of clear descriptions. See my comment below.

In your response to our comment about sport etc, you mention Carpersen et al. There is a reference to Casperen et al. That reference does not provide a definition/description of sport. Descriptions of what is considered sport, exercise and physical activity is still missing. Please be clear in your review what is considered sport, exercise and physical activity. In L392 is stated “Throughout the development of our research, we found it crucial to establish clear conceptual definitions of sport and exercise to accurately classify the RCT studies.” However, the reader will not find the “clear conceptual definitions” in your review.

R1. Dear Reviewer, we appreciate your comments, which have allowed us to greatly improve our manuscript. We highlight in light green, the changes requested by you.
We have added 2 new references, and incorporated the requested definitions between lines 129 and 138. We have also modified the line you indicate, now numbered as L409.

C2. Table 2. Are there books listed in the Table? Please reconsider the heading “journal/books”.

R2. We have changed the heading of table 2. Given the reported results, we have only used “journal”. (Line 219).